# Frailty in Community-Dwelling Adults Aged 40 Years and over with Type 2 Diabetes: Association with Self-Management Behaviors

**DOI:** 10.3390/ijerph19159092

**Published:** 2022-07-26

**Authors:** Zhijia Tang, Chunying Shen, Waikei Tong, Xiaoqiang Xiang, Zhen Feng, Bing Han

**Affiliations:** 1Department of Clinical Pharmacy and Pharmacy Administration, School of Pharmacy, Fudan University, Shanghai 201203, China; zjtang@fudan.edu.cn (Z.T.); 21211030001@m.fudan.edu.cn (W.T.); xiangxq@fudan.edu.cn (X.X.); 2Minhang Hospital & School of Pharmacy, Fudan University, Shanghai 201199, China; scysh@fudan.edu.cn

**Keywords:** diabetes mellitus, frailty, self-management, prevalence, community, risk factor

## Abstract

Background: Evidence is lacking on risk factors for frailty and prefrailty and their relationship with self-management behaviors in patients ≥40 years of age with type 2 diabetes. Methods: Participants were selected as a cross-sectional cohort at five communities in Shanghai, China during January–March 2021. The modified FRAIL scale and the Summary of Diabetes Self-Care Activities (SDSCA) measure were used. Results: Of the 558 participants, 10.2% were classified as frailty and 34.1% as prefrailty. The prevalence of frailty was higher in males than in females (*p* = 0.009), whereas females were associated with higher odds of prefrailty (aOR 1.67, 95% CI [1.08–2.60]). Multimorbidity, ≥3 chronic diseases, and hospitalization in the past year were considered risk factors for both frailty and prefrailty. Each point earned on SDSCA and physical activity were associated with lower odds of frailty (aOR 0.95, 95% CI [0.92–0.98]) and prefrailty (aOR 0.52, 95% CI [0.31–0.85]), respectively. Frail participants performed significantly worse self-care practice than prefrail and non-frail ones, especially on diet, physical activity, and medication adherence (*p* < 0.001). Conclusions: Frail patients ≥40 years of age with type 2 diabetes reported poorer self-care performance. Further interventional studies are warranted to clarify their causal relationship.

## 1. Introduction

Diabetes mellitus is a major public health issue across the world, which imposes a considerable socioeconomic burden worldwide [1]. China is now the world’s largest diabetes endemic country. An estimated 108 million people aged from 20 to 79 years in China had diabetes in 2021, by far the highest number of any country [2]. By 2045, the number is anticipated to reach 147 million [3]. The prevalence was even higher among the elderly at 22.5% [4]. Therefore, the care of patients with diabetes is of great importance with national-level prioritization.

Frailty is a status of decreased physiological reserve that is common among the middle-aged and elderly population [5]. Fried et al. described the most commonly used definition of frailty as the presence of three or more of the following characteristics: self-reported exhaustion, muscle weakness, slow walking speed, low physical activity, and unintentional weight loss [5,6]. Many studies have proven that frail people were at greater risk of disability, dependence, hospitalization and death [7,8,9,10]. Frailty is complex and multifaceted, often coexisting with other diseases, such as heart failure, that share common pathophysiological pathways and are associated with adverse outcomes that should be adequately assessed and comprehensively intervened [11,12]. Diabetes was considered one of the risk factors for frailty, the presence of which, in turn, was an important contributor to poor prognosis in older adults with diabetes [13,14]. Frail diabetic patients also appeared to have higher mortality rates than robust patients [15]. Although the prevalence of frailty increases with age, frailty can also affect younger people [16]. Studies have suggested that frailty was a preventable and reversible condition [6], particularly at an early stage [17,18], suggesting timely identification and management of frailty may be important for improving diabetes outcomes from the clinical point of view. Despite the potential importance of frailty in the middle-aged population, studies mostly relied on patients over 65 years old to complete the assessment. The prevalence of frailty and its subclinical state, prefrailty, in relatively young populations such as middle age remains to be determined. This is the first attempt to evaluate the frailty status of patients with diabetes dwelling in Chinese community, involving a much larger age range than previous studies.

Self-management behaviors refer to the actions taken by patients to deal with the disease based on their knowledge and skills [19]. This is especially vital for patients with chronic diseases such as hypertension and diabetes, because in addition to following their doctor’s orders, they must take personal responsibility toward their illnesses in daily lives. They are the real masters of their own health and well-being, and decide about whether to change lifestyles, exercise, eat healthy, and take medications [20]. Rigorous diabetes self-care practices have been shown to be effective in controlling blood glucose, preventing complications, and reducing long-term morbidity and mortality [21,22,23,24,25]. However, due to decline in self-control and memory, older adults often struggle to manage these tasks. They tend to cling to false beliefs and slip back into bad habits, making them particularly vulnerable to adverse health outcomes [26]. Their amount of self-care was significantly influenced by factors such as gender, education level, economic status, social support, and the duration of the disease [27]. On the other hand, although the evidence is scant, young adults with type 2 diabetes have been shown to perform worse than their older counterparts on some aspects of self-management, especially medication adherence [28,29]. Therefore, studies targeting younger patients are also warranted.

There is some evidence that high levels of self-care are associated with a lower incidence of frailty among older patients with cardiovascular conditions [30]. However, to the best of our knowledge, such correlation analysis is still lacking in diabetes. Given the rising prevalence of diabetes and the aging population in China, there is an urgent need to clarify it. The purpose of this study was to explore the prevalence and risk factors for frailty and prefrailty, and to evaluate the relationship between frailty and self-management behaviors in community-dwelling Chinese aged ≥40 years of age with diabetes.

## 2. Methods

### 2.1. Design, Setting, and Participants

This was a cross-sectional study and all participants were recruited through convenience sampling from the outpatient units of five community health centers in Shanghai, China, between January 2021 and March 2021. Experienced interviewers (i.e., general physicians, pharmacists, and nurses) administered questionnaires face-to-face to collect data on characteristics and medical history. Assistance (i.e., reading questions) was offered to participants who cannot fill out the questionnaire by themselves (i.e., having poor eyesight). The inclusion criteria were people ≥40 years of age, diagnosed with type 2 diabetes ≥3 months prior to the study, and able to provide informed consent. People with severe diabetes-unrelated organ damage, cognitive impairment, dementia, psychiatric disorders, and those unwilling to respond to the questionnaire were excluded. This study was approved by the Ethics Committee of Minhang Hospital (No. 2020-048-01K). Written informed consent was obtained from all participants.

### 2.2. Instruments for Data Collection

Frailty status was evaluated using a modified FRAIL scale in this study. The FRAIL scale is a time- and cost-effective frailty screening instrument with acceptable sensitivity and specificity compared to the commonly used screening tool, the frailty phenotype (FP) [5,31]. The scale contains five questions to assess the presence of fatigue, muscle resistance, ambulation, disease burden, and weight loss. Scoring is 0 for “no” and 1 for “yes”. Participants with a score ≥3 were classified as frail, those with a score of 1–2 as prefrail, and those with a score of 0 as non-frail [5].

The Chinese version of Summary of Diabetes Self-Care Activities (SDSCA) measure, a self-reported questionnaire with high reliability and validity, was employed to assess the frequency of performing self-care activities over the last seven days [32,33]. The SDSCA is one of the most common and widely used measures by clinicians [34]. The instrument consists of 12 items, covering six important domains of self-care practices: diet (general and special), physical activity, medication adherence, blood glucose monitoring, foot care, and smoking [33,35]. Except for a true-false question about smoking, each item scores on a scale of 0 to 7 with higher scores indicating better self-management. The total score of the questionnaire was 77. In order to help better describe the self-care performance, we defined a score ≥62 as “excellent”, 39–61 as “moderate”, and ≤38 as “poor” in this study. The Cronbach’s alpha coefficient (excluding the item on smoking) was 0.732, indicating acceptable internal consistency. The Kaiser-Meyer-Olkin value of 0.625 indicated an acceptable score with a significant Bartlett’s test of Sphericity (*p* < 0.001).

Copies of the modified FRAIL and SDSCA scale were distributed to diabetic patients who met inclusion criteria during outpatient encounters. The questionnaire survey was designed to obtain other self-reported information as well, including age, gender, height, weight, education level, socioeconomic, marital and cohabitation status, alcohol intake, smoking, duration of diabetes, medication use, and history of comorbidities and chronic diseases. Biochemical parameters such as HbA1c and fasting blood glucose (FBG) were directly retrieved from the hospital information system (HIS). All data were anonymized.

### 2.3. Statistical Analysis

The data were presented as number (percentage) or means (standard deviation). For categorical variables, the chi-square test and Fisher’s exact test were used to test the difference among non-frail, prefrail, and frail persons. For continuous variables, independent t-test and analysis of variance were used. A multinomial logistic regression model was used to compare sociodemographic characteristics and the SDSCA score of frail or pre-frail participants with non-frail participants. All significant variables in descriptive statistics were included in the final model. Adjusted odds ratios (aOR) and 95% confidence interval (CI) were calculated for the frail and prefrail groups, respectively, compared to the reference group (non-frail). The significance level was set at *p* < 0.05. All statistical analyses were performed by using the SPSS software, version 23 (IBM, Armonk, NY, USA).

## 3. Results

### 3.1. Characteristics of Study Sample and Prevalence of Frailty and Prefrailty

We included 558 participants with type 2 diabetes from the community. The majority (50.4%) were males, with a mean age of 69.1 ± 9.0 years. Most of them were unemployed or retired (91.4%), married (87.6%), and living with family or partners (95.0%); only 5.2% participants had a bachelor’s degree or higher. The mean duration of diabetes was 12.3 ± 7.8 years (Table 1).

The prevalence of frailty and prefrailty were 10.2% (*n* = 57) and 34.1% (*n* = 190), respectively. As shown in Table 1, the prevalence of frailty in men was more than twice that in women, but the prevalence of prefrailty was higher in women (*p* = 0.009). The proportion of frailty significantly increased with age, unemployment, HbA1c, number of comorbidities, chronic diseases and medications, and disease duration (*p* < 0.05). However, changes in the prevalence of prefrailty did not fully conform to this pattern. In addition, the frail group had markedly lower levels of physical activity, alcohol consumption, and hospitalization in the past year compared to other two groups.

### 3.2. Analysis of Self-Care Performance

Among the 558 participants, the mean SDSCA score was 39.03 ± 14.03, of which 54.8% (*n* = 306) had poor self-management, 36.2% (*n* = 202) had moderate self-management, and only 9.0% (*n* = 50) had excellent self-management. According to Table 2, participants were most likely to adhere to medication (5.34 ± 2.64), followed by diet (4.59 ± 1.51), and least likely to monitor blood glucose (1.91 ± 1.99). Compared with the prefrail and non-frail participants, the frail participants had a significantly lower mean SDSCA score of only 30.75 ± 8.60, indicating that their self-care performance was the least satisfactory, especially in diet, physical activity, and medication adherence (*p* < 0.001).

As shown in Table 3, females scored significantly higher than males (*p* = 0.018). In addition, those with higher education levels, lower HbA1c, and fewer comorbidities also had significantly higher SDSCA scores (*p* < 0.05). There were no differences when participants were stratified by any other variables.

### 3.3. Risk Factors for Frailty/Prefrailty and Their Relationship with Self-Management Behaviors

We then used multinomial logistic regression analysis to assess the relationship between frailty and self-management behaviors, as well as the risk of developing frailty and prefrailty (Table 4). Overall, patients with multimorbidity or ≥3 chronic diseases were more likely to be frail and prefrail compared to other groups, except for 1–2 vs. ≥3 chronic diseases that did not differ in the odds of prefrailty. Hospitalization in the past year was also of concern (frail: aOR 4.67, 95% CI [1.96–11.14]; prefrail: 4.27, [2.27–8.03], respectively). Female gender was considered another risk factor for prefrailty (aOR 1.67, 95% CI [1.08–2.60]). In contrast, physical activity worked as a protective factor for prefrailty (aOR 0.52, 95% CI [0.31–0.85]), and each point scored on the SDSCA scale was associated with lower odds of frailty (aOR 0.95, 95% CI [0.92–0.98]).

## 4. Discussion

In this cross-sectional study, we found that 10.2% and 34.1% of participants met the criteria for frailty and prefrailty, respectively, using the modified FRAIL scale. Although previous studies have noted that people with diabetes were more prone to frailty than those without diabetes, the reported prevalence in the diabetic population varied widely (ranging from 5% to 48%) [36,37]. In line with previous studies [38,39,40], our study showed that the prevalence of frailty significantly increased with age (*p* < 0.001). The mean age of our sample population was 69.1 ± 9.0 years. The correlation between frailty and age greatly heightened its importance in the context of an ageing global population [41]. Several studies also revealed that people with diabetes were more likely to become frail at a younger age [37,42]. In addition to age, Table 1 demonstrated that the prevalence of frailty also increased with unemployment, HbA1c, number of comorbidities, chronic diseases and medications, and disease duration (*p* < 0.05). These findings may help identify those at greatest risk and those most likely to benefit from optimization of treatment regimens [18,43].

In addition to the frail, individuals with prefrailty also accounted for a considerable proportion (34.1%) in this study. Since the transition between different frailty states was reversible, prefrailty, the subclinical phase of frailty, should be identified and intervened in as early as possible. The earlier screening and intervention begin, the greater the benefit to patients and healthcare systems [18]. Moreover, we noticed that the prevalence of prefrailty increased with age, unemployment, and number of comorbidities, chronic diseases, and medications such as frailty (Table 1). Previous studies revealed that risk factors such as older age, stroke, lower cognitive function, osteoarthritis, and hospitalizations were strongly associated with deterioration in frailty status among prefrail patients [42]. In support of this conclusion, our result showed that multimorbidity, ≥3 chronic diseases, and hospitalization in the past year were independent risk factors for both frailty and prefrailty (Table 4). An earlier meta-analysis of community-dwelling diabetic patients found that the pooled ORs for hospitalization due to frailty and prefrailty were 5.18 (95% CI 2.68–9.99) and 2.15 (95% CI 1.30–3.54), respectively [44], which was consistent with our results of 4.67 (95% CI 1.96–11.14) and 4.27 (95% CI 2.27–8.03). At the same time, although diabetes has been shown to be associated with fragility fractures in middle-aged men (relative risk [RR] 2.38, 95% CI [1.65–3.42]) and women (RR 1.87, 95% CI [1.26–2.79]) [45], in this study, being female was only considered a risk factor for prefrailty (aOR 1.67, 95% CI [1.08–2.60]), but not frailty (aOR 0.88, 95% CI [0.42–1.81]). Contrastingly, physical activity was related to lower odds of prefrailty (aOR 0.52, 95% CI [0.31–0.85]), which was also consistent with the literature [46,47].

In this study, the mean score of the SDSCA measure was 39.03 ± 14.03. Most of the participants had poor or moderate self-care performance; only 9.0% could be classified as excellent by our criteria. This finding was consistent with other research indicating that people with diabetes in many countries had poor self-care habits [48,49]. Not surprisingly, frail participants reported significantly worse self-care performance than those who were classified as prefrail and non-frail, especially in diet, physical activity, and medication adherence (*p* < 0.001, Table 2). This was also supported by the results of regression analysis shown in Table 4 (aOR 0.95, 95% CI [0.92–0.98]).

Of the five domains of SDSCA, BGM had the lowest mean score (1.91 ± 1.99), indicating it was the most difficult task for participants to accomplish. Similar findings have been documented [50,51,52,53]. Although self-management were already known to be particularly important for elderly diabetic patients, most Chinese patients did not perform BGM as recommended [54]. One explanation for this may be the cost of testing devices. The population included in our study had relatively low levels of education and employment, and may not recognize the importance of BGM nor afford the additional cost. However, despite having the lowest score, the frail and prefrail group showed a paradoxical, insignificant, but higher BGM score compared to the non-frail group (Table 2). This could be partly explained by the perceived support from family or cohabitants for handling testing tools.

Contrastingly, Gatt et al. [55] claimed that physical activity was the least self-care activity carried out by participants. The overall level of physical activity in this study was fine, probably because the rural population was more physically active on a daily basis, however it should be noted that frail participants scored significantly lower (*p* < 0.001). One possible reason for this contradiction is the inactivity of these patients caused by the COVID-19 quarantine and lockdown.

With regard to the domain where participants performed most satisfactorily, the findings varied considerably. Our results were consistent with most of the literature [50,52,53,55], which reported the highest score for medication taking, whereas Jackson et al. [51] declared that general diet was the most common self-care activity. Management of diabetes requires long-term compliance. However, according to our results, although the participants’ medication adherence was great overall with a highest mean SDSCA score of 5.34 ± 2.64, frail patients had poorer performance than other groups (*p* < 0.001). Similarly, although diet was the second-best self-management behavior overall (SDSCA score: 4.59 ± 1.51), frail patients had significantly worse eating habits than non-frail and prefrail ones (*p* < 0.001). Most participants reported that their meals were cooked at home. Frail people with difficulty getting out of the house were less likely to regularly purchase foods that are not easy to preserve, such as fruits or vegetables. This may be why the frail participants scored lower on diet. As shown in Table 3, sociodemographic and clinical characteristics including male gender, lower education level, higher HbA1c, and more comorbidities were significantly associated with poor self-care performance (*p* < 0.05). Similar findings have been observed in previous surveys [48,49,56,57]. This evidence provided references for individualized clinical, academic, and behavior interventions.

Our study assessed the frailty status and self-care performance in community-dwelling diabetic patients. To our knowledge, this is the first study to simultaneously explore the prevalence and risk factors of frailty and prefrailty in Chinese patients aged ≥40 years with type 2 diabetes, and to evaluate their association with self-management behaviors. On one hand, loss of self-efficacy may generally accelerate the transition to frailty [15]. The evidence is that diabetic patients with higher self-efficacy have demonstrated better self-management behaviors in diet, exercise, BGM, and taking medication [53]. On the other hand, frail patients may face great obstacles in performing self-care activities, for instance, disability, unexpected falls, fractures, worsening mobility, and cognitive decline [58]. Our findings suggested that good habits in diet, physical activity, and medication adherence may markedly help reduce possibility of frailty among diabetic patients, which should be emphasized in future management of diabetes. Our data were collected during the COVID-19 era when the public healthcare system was almost facing stagnation. Our results underscored the critical role of self-management behaviors in maintaining health and vitality in times of scarce community medical resources.

However, we do realize that our study has several limitations. First, the participants were selected from five community health centers in Shanghai by convenience sampling, which potentially reduced the representativeness of the sample population. Certain subgroups were underrepresented in this study, such as those with a bachelor’s degree or higher (only 5.2%), those who lived alone (5.0%), and those who received lifestyle interventions without taking any hypoglycemic drugs (3.4%). Thus, our findings should not be directly generalized to all patients with diabetes. Selection biases may also influence our conclusions as the participants may be more willing to engage in treatment and self-care than those who did not join the survey. Furthermore, as health authorities discouraged in-person visits during the COVID-19 pandemic, some eligible patients may be prevented from participating in the study, thus introducing sampling bias. Second, our findings were based on cross-sectional data. Therefore, causality could not be ascertained between frailty status and self-management behaviors. Although the Cronbach alpha coefficient was acceptable (0.732), the construct validity of the SDSCA measure used was not assessed considering the relatively low test-retest reliability [59]. Moreover, we neither recruited severely ill people nor compared the results to healthy people in this study, which may underestimate the prevalence of frailty as well as overestimate the overall self-care performance. Further prospective studies involving a larger and more general population are warranted.

## 5. Conclusions

This study provided epidemiological evidence for the prevalence of frailty and prefrailty among community-dwelling diabetic adults aged ≥40 years. Multiple sociodemographic and clinical variables have been proven to play key roles in the development of frailty and prefrailty. There was a significantly negative correlation between frailty and self-management behaviors. Further research should focus on identifying ways to enhance self-care activities in order to delay the onset of frailty, and clarifying possible causal relationships between them.

## Figures and Tables

**Table 1 ijerph-19-09092-t001:** Sociodemographic and clinical characteristics of participants, and prevalence of frailty and prefrailty in different subgroups (*n* = 558).

Characteristics		Frailty Status, *n* (%)				*p* Value
		All	Non-Frail (*n* = 311)	Prefrail (*n* = 190)	Frail (*n* = 57)	
Gender	Female	277 (49.6)	155 (56.0)	104 (37.5)	18 (6.5)	0.009
	Male	281 (50.4)	156 (55.5)	86 (30.6)	39 (13.9)	
Age (years)	40–65	135 (24.2)	99 (73.3)	31 (23.0)	5 (3.7)	<0.001
	65–75	278 (49.8)	146 (52.5)	100 (36.0)	32 (11.5)	
	>75	130 (23.3)	59 (45.4)	55 (42.3)	16 (12.3)	
Body mass index (kg/m^2^)	<25	337 (60.4)	187 (55.5)	112 (33.2)	38 (11.3)	0.424
	25–29.9	185 (33.2)	103 (55.7)	68 (36.8)	14 (7.6)	
	≥30	23 (4.1)	13 (56.5)	6 (26.1)	4 (17.4)	
Education level	Secondary or lower	283 (50.7)	155 (54.8)	96 (33.9)	32 (11.3)	0.129
	High school/associate	246 (44.1)	133 (54.1)	89 (36.2)	24 (9.8)	
	Bachelor or over	29 (5.2)	23 (79.3)	5 (17.2)	1 (3.4)	
Socioeconomic status	Employed/self-employed	48 (8.6)	40 (83.3)	8 (16.7)	0 (0)	<0.001
	Unemployed/retired	510 (91.4)	271 (53.1)	182 (35.7)	57 (11.2)	
Marital status	Single/divorced/widowed	69 (12.4)	31 (44.9)	31 (44.9)	7 (10.1)	0.112
	Married	489 (87.6)	280 (57.3)	159 (32.5)	50 (10.2)	
Cohabitation status	Solitude	28 (5.0)	13 (46.4)	12 (42.9)	3 (10.7)	0.530
	Cohabitated	530 (95.0)	298 (56.2)	178 (33.6)	54 (10.2)	
HbA1c (%)	≤6.5	120 (21.5)	71 (59.2)	46 (38.3)	3 (2.5)	0.003
	>6.5	394 (70.6)	205 (52.0)	135 (34.3)	54 (13.7)	
FBG (mmol/L)	≤7.0	319 (57.2)	174 (54.5)	107 (33.5)	38 (11.9)	0.299
	>7.0	229 (41.0)	129 (56.3)	82 (35.8)	18 (7.9)	
No. of comorbidities	0	300 (53.8)	211 (70.3)	79 (26.3)	10 (3.3)	<0.001
	1–2	177 (31.7)	86 (48.6)	71 (40.1)	20 (11.3)	
	≥3	81 (14.5)	14 (17.3)	40 (49.4)	27 (33.3)	
No. of chronic diseases	0	145 (26.0)	107 (73.8)	32 (22.1)	6 (4.1)	<0.001
	1–2	363 (65.1)	196 (54.0)	133 (36.6)	34 (9.4)	
	≥3	50 (9.0)	8 (16.0)	25 (50.0)	17 (34.0)	
No. of medications	0	19 (3.4)	14 (73.7)	4 (21.1)	1 (5.3)	0.001
	1–2	239 (42.8)	157 (65.7)	72 (30.1)	10 (4.2)	
	≥3	79 (14.2)	33 (41.8)	37 (46.8)	9 (11.4)	
Duration of diabetes (years)	<10	237 (42.5)	147 (62.0)	76 (32.1)	14 (5.9)	0.001
	10–20	213 (38.2)	111 (52.1)	79 (37.1)	23 (10.8)	
	>20	98 (17.6)	45 (45.9)	33 (33.7)	20 (20.4)	
Physical activity, yes		146 (26.2)	104 (71.2)	37 (25.3)	5 (3.4)	<0.001
Smoking, yes		73 (13.1)	48 (65.8)	20 (27.4)	5 (6.8)	0.171
Regular alcohol consumption, yes		40 (86.9)	27 (67.5)	13 (32.5)	0 (0)	0.033
Cardiovascular disease, yes		348 (62.4)	185 (53.2)	122 (35.1)	41 (11.8)	0.166
Hospitalization in the past year, yes		89 (15.9)	23 (25.8)	50 (56.2)	16 (18.0)	<0.001
Emergency department visits in the past year, yes		34 (6.1)	14 (41.2)	13 (38.2)	7 (20.6)	0.072

Abbreviation: FBG fasting blood glucose.

**Table 2 ijerph-19-09092-t002:** Scores of the Summary of Diabetes Self-Care Activities (SDSCA) measure of participants (*n* = 558).

Domains	No. of Items	Mean Score ± SD				*p* Value
		All	Frail (*n* = 57)	Prefrail (*n* = 190)	Non-Frail (*n* = 311)	
Diet	4	4.59 ± 1.51	3.71 ± 1.64	4.57 ± 1.52	4.77 ± 1.42	<0.001
Physical activity	2	3.08 ± 2.08	1.92 ± 1.68	2.97 ± 2.18	3.36 ± 2.00	<0.001
Blood glucose monitoring	2	1.91 ± 1.99	1.93 ± 1.43	2.02 ± 2.02	1.84 ± 2.07	0.611
Foot care	2	2.72 ± 2.75	2.11 ± 1.78	2.72 ± 2.75	2.84 ± 2.88	0.186
Medication adherence	1	5.34 ± 2.64	4.11 ± 2.51	5.68 ± 2.30	5.36 ± 2.80	<0.001
Total	12	39.03 ± 14.03	30.75 ± 8.60	39.27 ± 15.20	40.40 ± 13.60	<0.001

**Table 3 ijerph-19-09092-t003:** Univariate analysis of the Summary of Diabetes Self-Care Activities (SDSCA) measure stratified by characteristics.

Variables		Mean Score	*p* Value
Gender	Female	40.45 ± 13.80	0.018
	Male	37.63 ± 14.14	
Age (years)	40–64	37.90 ± 12.35	0.273
	65–75	40.09 ± 14.96	
	>75	38.38 ± 13.92	
Body mass index (kg/m^2^)	<25	39.17 ± 13.83	0.951
	25–29.9	39.18 ± 14.45	
	≥30	38.22 ± 14.86	
Education level	Secondary or lower	36.75 ± 12.87	<0.001
	High school/associate/bachelor or over	41.38 ± 14.79	
Socioeconomic status	Employed/self-employed	35.90 ± 11.68	0.061
	Unemployed/retired	39.33 ± 14.20	
Marital status	Single/divorced/widowed	39.42 ± 12.56	0.805
	Married	38.98 ± 14.23	
Cohabitation status	Solitude	38.86 ± 14.19	0.947
	Cohabitated	39.04 ± 14.03	
HbA1c (%)	≤6.5	42.70 ± 16.57	0.002
	>6.5	37.44 ± 12.72	
FBG (mmol/L)	≤7.0	39.03 ± 14.05	0.941
	>7.0	39.12 ± 14.17	
No. of comorbidities	0	41.07 ± 14.41	<0.001
	1–2	37.82 ± 13.55	
	≥3	34.12 ± 12.10	
No. of chronic diseases	0	40.18 ± 12.45	0.074
	1–2	39.13 ± 14.80	
	≥3	34.96 ± 11.96	
No. of medications	0	34.16 ± 9.29	0.156
	1–2	40.64 ± 15.11	
	≥3	41.10 ± 13.78	
Duration of diabetes (years)	<10	40.19 ± 15.10	0.161
	10–20	37.77 ± 13.56	
	>20	38.12 ± 12.31	
Smoking, yes		37.84 ± 15.42	0.436
Regular alcohol consumption, yes		40.40 ± 14.58	0.540
Cardiovascular disease, yes		38.35 ± 14.16	0.142
Hospitalization in the past year, yes		37.78 ± 14.63	0.358
Emergency department visit in the past year, yes		40.32 ± 14.03	0.580

Abbreviation: FBG fasting blood glucose.

**Table 4 ijerph-19-09092-t004:** Odds ratios from multinomial logistic regression for frailty and prefrailty.

Variables		aOR (95% CI)	
		Frail vs. Non-Frail	Prefrail vs. Non-Frail
Gender	Female	0.88 (0.42–1.81)	**1.67 (1.08–2.60)**
	Male	1 (reference)	1 (reference)
No. of comorbidities	0	**0.10 (0.04–0.29)**	**0.18 (0.09–0.39)**
	1–2	**0.24 (0.10–0.63)**	**0.36 (0.17–0.75)**
	≥3	1 (reference)	1 (reference)
No. of chronic diseases	0	**0.18 (0.04–0.72)**	**0.29 (0.11–0.80)**
	1–2	**0.25 (0.08–0.73)**	0.43 (0.17–1.07)
	≥3	1 (reference)	1 (reference)
Physical activity, yes		0.42 (0.14–1.24)	**0.52 (0.31–0.85)**
Hospitalization in the past year, yes		**4.67 (1.96–11.14)**	**4.27 (2.27–8.03)**
SDSCA score		**0.95 (0.92–0.98)**	1.00 (0.99–1.02)

The reference category was the non-frail group. Results were presented as adjusted OR (95% CI). The model was adjusted for gender, age, employment status, HbA1c, number of comorbidities, chronic diseases and medications, duration of diabetes, physical activity, history of hospitalization (categorical), and SDSCA scale score (continuous). Numbers in bold indicated significant findings. Abbreviations: aOR adjusted odds ratio; CI confidence interval; SDSCA Summary of Diabetes Self-Care Activities.

## Data Availability

Data associated with the findings of this study are included in the article and are available from the author upon reasonable request.

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
