# Peer review of "Frailty in Community-Dwelling Adults Aged 40 Years and over with Type 2 Diabetes: Association with Self-Management Behaviors"

_ijerph, 2022, doi:10.3390/ijerph19159092_

Round 1

Reviewer 1 Report

Dear Authors,

The purpose of this study was to explore the prevalence and risk factors for frailty and prefrailty, and to evaluate the relationship between frailty and self-management behaviors in community-dwelling Chinese aged ≥40 years of age with diabetes.

The study is of scientific interest and in line with the aims of the journal. The author guidelines have been respected and the work is well written.

Minor revisions:

Abstract

-       The abstract should be a total of about 200 words maximum. The abstract should be a single paragraph and should follow the style of structured abstracts, but without headings: 1) Background: Place the question addressed in a broad context and highlight the purpose of the study; 2) Methods: Describe briefly the main methods or treatments applied. Include any relevant preregistration numbers, and species and strains of any animals used. 3) Results: Summarize the article's main findings; and 4) Conclusion: Indicate the main conclusions or interpretations. The abstract should be an objective representation of the article: it must not contain results which are not presented and substantiated in the main text and should not exaggerate the main conclusions. (https://www.mdpi.com/journal/ijerph/instructions

Introduction

-       Frailty is a complex and multifaceted public health issue highly prevalent in older adults leading to increased direct and direct sanitary. In the Introduction section i suggest to better describe frailty condition,  reporting the most common overlapping features and disabling sequelae. (Please cite: 10.3390/nu14050982).

Methods

-       “This was a cross-sectional study that all participants were randomly recruited from 78 the outpatient units of five community health centers in Shanghai, China between January 79 and March 2021”. Please reformulate, it is not clear.

-       Was the study performed in accordance with the “Strengthening the Reporting of Observational Studies in Epidemiology” (STROBE) Guidelines?

The topic is very interesting and actual, and the work is well written. 

In my opinion, the manuscript is suitable for publication in this Journal after minor revision.

Reviewer 2 Report

1. Abstract. "Our findings suggested that enhancing self-management skills may reduce 27 the development of frailty..." Since the study is cross-sectional, this conclusion may mislead readers. Please revise the conclusion and avoid using words that imply causality. 

2. Introduction. In the first paragraph, the authors talked about young adults. However, in the second paragraph, they talked about self-management and the issues of self-control and memory in older adults. Which group is the main target? It seems the young adult group is the study sample. In this case, please revise the second paragraph and provide the self-care issues in young adults.

3. Introduction. If the authors wanted to talk about COVID-19, please provide more explanation about the relationship between diabetes and COVID-19. It seems that COVID-19 is irrelevant to the study topic.

4. Methods. It is not clear how the authors determined the cut-off scores for the SDSCA. Typically, cut-off scores are calculated by ROC analysis against gold-standard measures. Please clarify this issue. 

5. Methods. Instead of just providing the numeric psychometric properties of the SDSCA, please explain about meaning of the psychometric properties (e.g., moderate reliability, good construct validity from factor analysis, etc.).

6. Analysis. It is not clear why multinominal logistic regression was chosen over ordinal logistic regression since frailty has a hierarchical structure. Please clarify this issue. 

7. Analysis. "Based on the univariate analysis results..." It is not clear what the sentence means.

8. Analysis. The study utilized regression models to account for sociodemographic and clinical characteristics; however, also used simple unadjusted bivariate analysis. If the authors wanted to examine the association between the modified FRAIL and SDSCA scale, it would be better to include the SDSCA scale score in the regression model.

Reviewer 3 Report

Interesting topic chosen for research. The assessment of fragility should also be continued at a young age depending on different risk factors.  I think it could be accepted maybe with 2 minor modification :

1.The structure of the abstract- missing Material and method 2.They must provide more information about  the SDSCA.

Reviewer 4 Report

Manuscript shows the relation between frailty and diabetes type 2 in people older than 40 years. I have some considerations.

It would be convenient to add some reference on prevalence more updated than 2013 or 2018 at the introduction section...

Line 43: "many studies..." but you only mention one (7).

Methodology:

Were other pathologies or comorbidities taken into account as exclusion criteria?

There is a lack of information: Who collects the sample, when? How are the patients contacted? Are the patients admitted, or do they only attend consultations? Are they helped to fill it in, do they fill it in there or at home?

Which professionals are in charge of this study? Healthcare professionals, epidemiologists...

Further explanation of the habits of the present sample should be discussed.

The fact that it was carried out in Covid-19 period may be a bias to be considered in relation to the possible inactivity of these patients.

It would be interesting the addition of a control group, to establish differences.

Do they receive habits education at any time since the diagnosis?

References do not meet journal standards.

Round 2

Reviewer 2 Report

Thank you for addressing the previous comments.

Author Response

We appreciate the time and effort that you dedicated to providing feedback on our manuscript and are grateful for the valuable improvements to our paper. 

Reviewer 4 Report

Dear authors,

Thank you for taking into account my considerations.

Just one more thing, please add the explanation of the abbreviations used in the tables at the end of the tables.

Author Response

We appreciate the time and effort that you dedicated to providing feedback on our manuscript and are grateful for the insightful comments on our paper. In the newly submitted manuscript, the explanation of the abbreviations was added at the end of the tables.